# Multi-Layered Bipolar Ionic Diode Working in Broad Range Ion Concentration

**DOI:** 10.3390/mi14071311

**Published:** 2023-06-26

**Authors:** Jaehyun Kim, Cong Wang, Jungyul Park

**Affiliations:** 1Department of Mechanical Engineering, Sogang University, Sinsu-dong, Mapo-gu, Seoul 121-742, Republic of Korea; ssamdolr@naver.com; 2School of Mechanical Engineering and Electronic Information, China University of Geosciences (Wuhan), 388, Lumo Road, Wuhan 430074, China; congwang@cug.edu.cn

**Keywords:** ion current rectification, multi-layer, bipolar ionic diode, nanochannel network membrane, nanoparticles, hysteresis loop

## Abstract

Ion current rectification (ICR) is the ratio of ion current by forward bias to backward bias and is a critical indicator of diode performance. In previous studies, there have been many attempts to improve the performance of this ICR, but there is the intrinsic problem for geometric changes that induce ionic rectification due to fabrication problems. Additionally, the high ICR could be achieved in the narrow salt concentration range only. Here, we propose a multi-layered bipolar ionic diode based on an asymmetric nanochannel network membrane (NCNM), which is realized by soft lithography and self-assembly of homogenous-sized nanoparticles. Owing to the freely changeable geometry based on soft lithography, the ICR performance can be explored according to the variation of microchannel shape. The presented diode with multi-layered configuration shows strong ICR performance, and in a broad range of salt concentrations (0.1 mM~100 mM), steady ICR performance. It is interesting to note that when each anion-selective (AS) and cation-selective (CS) NCNM volume was similar to each optimized volume in a single-layered device, the maximum ICR was obtained. Multi-physics simulation, which reveals greater ionic concentration at the bipolar diode junction under forward bias and less depletion under backward in comparison to the single-layer scenario, supports this tendency as well. Additionally, under different frequencies and salt concentrations, a large-area hysteresis loop emerges, which indicates fascinating potential for electroosmotic pumps, memristors, biosensors, etc.

## 1. Introduction

Selective permeation of ions in cell membranes, such as the conduction of nerve impulses, play an important role in human activities. Research using artificial nano-channels has been extensively conducted, which mimics the ion-selective permeation of these cells. Artificial nano-channels are more sturdy and reliable than biological nanochannels and can be applied as energy harvesting [1,2,3,4,5,6,7,8,9,10], ionic memristors [11,12], an electroosmosis pump [13], etc. By introducing asymmetric properties to the nanochannel, the first nanofluid diode device was proposed by Chang et al. [14]. As the EDL in the conical nanopipette overlaps strongly toward the tip, it has high ion selectivity and counter-ions with opposite charge characteristics to the surface of the nanopipette pass smoothly. Therefore, ionic current flows from the base to the tip under the forward bias and is inhibited under the backward bias. After this work, various types of ionic diodes were introduced, and generally, ionic diodes can be divided into unipolar and bipolar diodes, depending on whether the polarity of the surface of the nanochannel is unipolar or bipolar [15]. Ion current rectification (ICR) [16,17,18] is a critical indicator of diode performance, and the ICR of bipolar ionic diodes is better than that of unipolar diodes. Typically, many proposed ionic diodes rely on nanochannels made through ion tract etching [19,20,21] and anodizing [22,23]. Nanoscale geometry control to control the degree of electrical double layer (EDL) overlapping or surface modification is used to induce the asymmetric ion transport and ionic current rectification. Various nanoscale shapes were introduced including symmetric geometry (cylindrical [24]) and asymmetric geometry (bullet-shaped [25], hourglass [26,27], dumbbell [28], funnel-shaped [29,30,31], trumpet, and cigarette [32]), and the numerical simulation for describing the ionic transfer behavior in the asymmetric nanoscale [9,32] and ion selectivity in single channel [33], and their applications for energy harvesting [1,2,3,4,5,6,7,8,9,10] were presented. Even so, there is intrinsic limitation for realizing arbitrary shapes on the nanoscale due to the complexity of fabrication steps and requirements of the specific equipment such as heavy ion sources.

In contrast to these studies, based on nanoscale geometry control, the nanochannel network membrane (NCNM) made of uniformly sized self-assembled nanoparticles can control asymmetric ion transport by changing the geometry of the micro-channel through a soft lithography process that is quick, inexpensive, and simple. Most importantly, almost free shaping is possible. In view of the nanoscale, uniform geometry (nanopores of uniform size generated by self-assembly of nanoparticles) is used. Nevertheless, the selective and directed ion transport can be regulated by microscale geometry control; smaller width shows higher ion selectivity and higher surface charge density [34]. Another differentiated point of the NCNM is that the surface charge density and polarity can be controllable by selecting the proper materials for nanoparticles.

Ionic diodes are normally accumulated in the forward bias and depleted in the reverse bias, which means that conductance increases in the forward bias and drops in the reverse bias. Due to this property, when an alternate (AC) potential is given to an ionic diode, history-dependent conductance or hysteresis in ionic conductance can develop. This phenomenon can be applied to an alternating current osmotic pump or an ionic memristor device. A memristor is a device that combines resistance and memory to carry out neuromorphic functions. Recently, a fluidic-based memristor has become highly desirable for mimicking biological synapses in a solution-based context due to its higher compatibility with biological systems and dealing with ions and chemical species [12,35]. The hysteresis loop is a key feature of memristors. In general, the greater the hysteresis width, the greater the amount of information that can be stored and the more stable it is. That is, if the large hysteresis width is secured in a broad range of salt concentrations, stable neuromorphic functions of a memristor can be expected.

In the previous study [36], we conducted the optimization for the parameters contributing to the performances in the single-layered bipolar ionic diode based on NCNM: nanoparticle size, the geometry of NCNMs, and concentration of electrolyte. By thorough investigation of these parameters, a high-rectification ratio was successfully achieved; however, this high performance was possible in only moderately low-concentrated electrolytes (1–10 mM). The ICR was degraded sharply in a very low (0.1 mM) or high (>100 mM) concentrated environment. Here, we propose a multi-layered bipolar ionic diode with different heights of each NCNM region, which is expected to secure the high-rectification ratio and maintain performance in the broad range of salt concentrations. The rectification characteristics and performance were investigated experimentally according to the variation of geometry in multi-layered microchannel and electrolyte concentration. Then, numerical simulations at 1, 10, and 100 mM for single and multi-layered were conducted and compared to support the experimental results. Finally, we also explored hysteresis loops by applying AC potential at the 1, 10, and 100 mM KCl for showing the potential as a memristor.

## 2. Materials and Methods

### 2.1. Fabrication Process for Multi-Layered Bipolar Ionic Diode

Microchannels for a multi-layered bipolar ionic diode were fabricated by a soft-lithography process method (Figure 1A). The negative photoresist was applied to the silicon wafer (PR, SU-8 2005; Microchem Co., Westborough, MA, USA) by using a spin coater and then soft-baked. Using the mask aligner equipment, the PR on the silicon wafer was exposed to UV to make a pattern, and the wafer was hard-baked. By removing the unexposed PR, a pattern of 1st shallow channel with a height of 5 µm was formed. Following, a 2nd shallow channel with a height of 25 µm, and a deep channel with a height of 100 µm were fabricated by using negative photoresist: SU8-2025 and SU8-3050 (Microchem Co., Westborough, MA, USA), respectively, with the same process in the above. When the master mold was complete, the surface was treated with (3,3,3-trifluoropropyl)silane (452807; Sigma-Aldrich, St. Louis, MO, USA). Then, polydimethylsiloxane (PDMS; Sylgard; Dow Corning Korea Ltd., Gwangju-si, Gyeonggi-do, Republic of Korea) was poured over the master mold and heated 95 °C for 1 h on a hot plate. The reservoir of the PDMS device was punched out with a 1.5 mm medical punch. The surface of PDMS and slide glass were treated by using plasma equipment (Cute-MP; Femto Science, Hwaseong-si, Gyeonggi-do, Republic of Korea) and attached to each other. AS-NCNM (anion selective-nanochannel network membrane) and CS-NCNM (cation selective-nanochannel network membrane) were fabricated by the self-assembly of NR_3_^+^ functionalized nanoparticles (Micromod, Rostock, Germany) and COOH^-^ functionalized nanoparticles (Micromod, Rostock, Germany). The multi-layered bipolar ionic diode was developed by constructing AS-NCNM and CS-NCNM at the designed region in the multi-layered microchannels. As shown in Figure 1B, ethanol 70% was injected though a reservoir in deep channel A. Ethanol was stopped at the boundary between shallow channel and deep channel B by neck pressure [37]. Following, diluted nanoparticles (surface: NR_3_^+^, Diameter: 200 nm) were injected in the deep channel and the nanoparticles moved to the shallows [38]. When the nanoparticles were filled in the desired region and the diluted nanoparticles were removed, AS-NCNM was completed. After diluted nanoparticle injection (surface: COOH^-^, diameter:100 nm) in the deep channel A, the nanoparticles were fully filled in the shallow channel. Then, the diluted nanoparticles were removed, and CS-NCNM was completed. The finished device was dried for one day and then used for the experiment.

### 2.2. Measuring Ion Current of Multi-Layered Bipolar Ionic Diode

Before the experiment, nanopores of NCNMs were washed twice with 70% ethanol and then washed three times with DI water. Subsequently, the target concentration of KCl was injected, waiting ~30 min for equilibrium to be reached. Platinum (PT) electrodes were connected into the reservoir of the PDMS device, applying voltage to the tip and grounding the base. The ion current was measured from −9 V to 9 V (interval 0.2 V per 1 s) using a picoammeter (Keithley 6487, Tektronix, Beaverton, OR, USA).

### 2.3. Numerical Modeling for Simulation

Modeling for the simulation of single- and multi-layers were performed using the finite element method (COMSOL, Multiphysics 5.6). The ion transport under the electric field was solved using “Transport of Diluted Species” and “Electrostatics” modules, based on Nernst Planck equation and Poisson’s equation, respectively [34,36]. The conservative equation for ion flux and the Poisson equation are defined as follows:(1)$$\frac{\partial c_{i}}{\partial t}=-\nabla \cdot J_{i}$$
(2)$$\nabla^{2}\varphi =-\frac{F}{\varepsilon_{0}\varepsilon_{r}}\sum z_{i}c_{i}$$
where *J_i_*, *φ*, *F*, and *z_i_* are the ion flux of species *i*, electrical potential, Faraday constant, and valence of species *i*, respectively. *c*, $\varepsilon_{0}$, and $\varepsilon_{r}$ are the ion concentration of ion species *i*, the vacuum permittivity, and the dielectric constant of the electrolyte solution. Following, the ion flux of each species is solved by Nernst–Planck Equation (3):(3)$$J_{i}=D_{i}\left(\nabla c_{i}+\frac{Fz_{i}c_{i}}{RT}\nabla \varphi \right)$$
where *D*, *R*, and *T* are coefficients of ions diffusion, constant of the gas universal, and solution temperature, respectively. Convection flow was ignored to focus on the comparison of rectification performance with respect to the difference in geometry between the single and multi-layer.

The simulation model was constructed with the same size as the real bipolar ionic device and the height of the channel was reflected using the out-of-plane thickness setting in the electrostatic module.

The volumetric free charge density in the deep channel is used by Equation (4):(4)$$\rho_{deep}=F\left(z_{cation}c_{cation}+z_{anion}c_{anion}\right)$$
where *z_cation_* and *z_anion_* are the charge numbers of cation and anion, and *c_cation_* and *c_anion_* are ion concentration of cation and anion, respectively. The volumetric free charge density in the shallow channel is used by Equation (6) with the *z_membrane_ c_membrane_* term as an assumption for the membrane on which nanoparticles are assembled.
(5)$$\rho_{shallow}=F\left(z_{cation}c_{cation}+z_{anion}c_{anion}+z_{membrane}c_{membrane}\right)$$
where *z_membrane_* and *c_membrane_* are the charge number of membrane surface and concentration of membrane. *c_membrane_* is calculated by Equation (6):(6)$$c_{membrane}=\frac{4\pi r^{2}n\sigma }{eN_{A}\left(1-0.74\right)V}$$
where *σ* is the surface charge density of silica nanoparticles, *n* is total number of nanoparticles in NCNM ($n=\frac{3}{4}\times 0.74\times \frac{V}{\pi r^{3}}$) [34,36], and *r*, *e*, and *N_A_* are nanoparticles radius, the electron charge, and Avogadro’s number, respectively.

### 2.4. Ion Current Measurement for Hysteresis

We used an AC voltage to draw a memristor hysteresis loop and a function generator (AFG3022B, Tektronix, Beaverton, OR, USA.) to apply the AC voltage and a picoammweter (6487, Keithley) to measure the ion current. AC voltages were used ranging from −5 V to 5 V; frequencies of 10, 1, and 0.1 Hz; and concentrations of 1, 10, and 100 mM. The order of injecting the solutions and connecting platinum (PT) electrodes was the same as in Section 2.2. The hysteresis loop is plotted for five periods with the gray line from −5 V to 5 V, and the bold black line is the average of the five period ion currents.

## 3. Results and Discussion

### 3.1. Working Principle of Multi-Layered Bipolar Ionic Diode Based on NCNM

Figure 2A shows a schematic of a multi-layered bipolar ionic diode based on NCNM with ion distributions due to its polarized surface charge and asymmetric geometry. Figure 2B depicts the image of an inverted microscope of the bipolar diode according to the proposed fabrication process. Figure 2C shows the ionic transport and distributions rely on the applied potential. The bipolar diode has the preferred ionic distributions in terms of surface charge polarity. A heterogeneous ionic junction, analogous to a p–n junction in a solid-state diode device, forms at the interface of two oppositely charged surfaces. The ions are in equilibrium without potential bias; cations are in CS-NCNM, and anions are in AS-NCNM due to electrostatic interactions between the ions and surface charge. When a forward bias is applied, an accumulation region near the heterogeneous junction forms in which both ions are concentrated, and potential gradients along the NCNM drive the high transport of ions. When the bias is reversed, a depletion region forms at the junction, and ion transport is suppressed.

### 3.2. Characterization of Ionic Rectification According to the Change of Geometry

Ionic rectification was investigated with respect to the change of geometry in NCNMs experimentally. As shown in Figure 3A, the cuboids (width: 200 µm, height: 100 µm, length: 150 µm) are referred to as reference volumes, and the volume ratio of AS-NCNM and CS-NCNM to the reference volume is called ANR and CNR, respectively. The initial geometry parameters were based on the optimized geometry result from our previous work for a single-layered case [36], which has CS-NCNM filled with negatively charged particles (diameter: 100 nm, surface: COOH^−^) and AS-NCNM filled with positively charged particles (diameter: 200 nm, surface: NR_3_^+^). From this, we increased the height of AS-NCNM from 5 µm to 25 µm and changed the width of AS-NCNM to 40 µm, 100 µm, and 200 µm, respectively. The smaller the volume ratio of AS-NCNM under a KCl concentration of 10 mM, the higher the ICR (ICR 154 at CNR: 0.0025, ANR: 0.0833, ICR 243 at CNR: 0.0025, ANR: 0.0417, ICR 367 at CNR: 0.0025, ANR: 0.0167). This result is supported by our previous result where the smaller area shows higher ion selectivity [34].

Following, we performed a similar investigation for observing the effect of the change in the length of CS-NCNM and AS-NCNM (Figure 4A). Overall, the higher the total volume ratio of NCNM, the higher the ion current. The maximum ICR value was found when the CNR to ANR volume ratio was 0.0025:0.0167, and very surprisingly, the volume for AS-NCNM and CS-NCNM in the optimal geometry in the multi-layered device was similar to the volumes for AS-NCNM and CS-NCNM in the single-layered device [36].

### 3.3. KCl Concentration Optimization of the Multi-Layered Bipolar Ionic Diode

The ratio between the nanopore size and EDL thickness (R = d_n_/λ_D_, λ_D_, and d_n_ are the EDL and nanopore size, respectively) is a critical factor for ion selectivity at NCNMs, and naturally, the factor R is also affected by the EDL thickness related with the ion concentration [34]. Therefore, an experiment was conducted to increase the ICR by adjusting the concentration of KCl to 0.1, 1, 3, 10, 100, and 400 mM. As a result, it was optimized at 3 mM as ICR ~626. Especially in KCl 0.1 mM and 100 mM (Figure 4C), the ICR value of multi-layered bipolar ionic diode is, respectively, 1.5-times and 25-times better than single-layered bipolar ionic diode in the same concentration. As shown in the numerical simulation results (Figure 5), when KCl 100 mM was applied to a single-layered ionic diode, KCl was accumulated four times (400 mM) for forward bias, but five times (500 mM) for multi-layered. In addition, the ion depletion phenomena of multi-layered was slightly higher than that of single-layered in backward bias. In Figure 5C, the accumulation at 1 mM was 20 times in the single-layered and 40 times in the multi-layered, resulting in the multi-layered working better. In other words, the ionic rectification phenomenon of multi-layered was more prominent at higher and lower concentrations than that of single layers. As shown in Table 1, many bipolar diodes, including our single-layered case, have a high rectification ratio in the specific salt concentration (although some cases have higher rectification than ours), but the proposed multi-layered device has stable rectification performance over a wide range of salt concentrations.

### 3.4. Hysteresis Loop of Multi-Layered Bipolar Ionic Diode

From the Section 3.3 results, we found that multi-layered bipolar ionic diodes operate reliably over a wide range of salt concentrations. High performance of ion rectification means that there is a distinction between ion accumulation and depletion phenomenon depending on the bias direction. When ions are accumulated, the conductivity of the multi-layered bipolar ionic diode increases, and when ions are depleted, the conductivity decreases. As shown in Figure 6B–D, when an AC potential ranging from −5 V to +5 V is applied, the multi-layered bipolar ionic diode has high current values at positive voltages and low current values at negative voltages. It has been reported that a hysteretic loop is drawn in the current–voltage diagram when a periodic voltage is applied to a conical nanochannel (as memristor) [11,39]. We conducted experiments under 10, 1, and 0.1 Hz conditions and found hysteresis loops that crossed zero at frequencies of 1 and 0.1 Hz, and it had the optimal hysteresis loop at 0.1 Hz at KCl 10 mM, as shown in Figure 6A. The width of the hysteresis is an important indicator of the memory capacity and stability of the memristor device, and the larger the width of the hysteresis, the more information storage or signal processing functions can be realized. The broad hysteresis with these characteristics was at KCl 10 mM and 0.1 Hz. In addition, the zero-crossing wide hysteresis loop in ionic memristors has been studied at the salt concentration of 10 mM, normally [12,40]. Figure 6B–D shows the hysteresis loop at KCl 1 mM, 10 mM, and 100 mM, respectively, and the best form of hysteresis loop was observed at 10 mM. The multi-layered bipolar ionic diode has decent hysteresis loops at lower (KCl 1 mM) and higher concentrations (KCl 100 mM) also. This shows the feasibility of bipolar diodes as memristors, especially in multi-layered bipolar ionic diodes, which operate over a wide range of concentrations.

## 4. Conclusions

We propose a multi-layered bipolar ionic diode made of AS- and CS-NCNM produced by nanoparticle self-assembly in this study. The geometry modification of NCNM and surface charge management is very simple and freely adjustable due to the relatively simple soft-lithography and functionalization, proper size, and material selection of nanoparticles. ICR features were explored with respect to changes in NCNM shape and ion concentration, and we discovered that the highest ICR in a multi-layered ionic diode was secured when each AS- and CS-NCNM volume was the same as each volume in a single-layered device (our previous result). The multi-layered diode exhibited stable ICR performance over a wide range of salt concentrations. Additionally, the outcomes of the numerical simulations showed that the multi-layered bipolar ionic diode performs ion rectification better than the single-layered device in terms of overall salt concentration. Furthermore, by applying an alternating current potential to a multi-layered bipolar ionic diode, an ordinary hysteresis loop for the memristor was created, and decent hysteresis loops were found over a wide range of salt concentrations. This demonstrates the promising potential of the suggested ionic diodes as memristors, electroosmotic pumps, biosensing, and energy harvesting devices that work at low and high concentrations consistently. In terms of geometry change, other shape changes can be explored in the near future to improve the performance of ionic diodes such as tapered with various angles, hyperbolic shape, etc.

## Figures and Tables

**Figure 1 micromachines-14-01311-f001:**
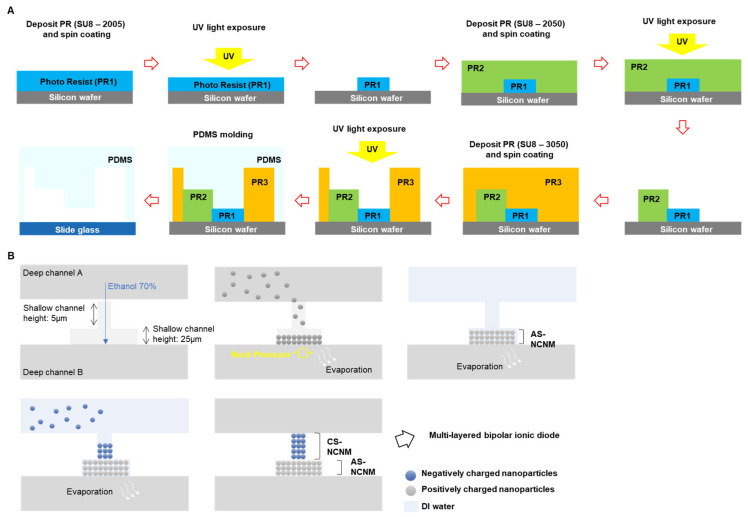
Fabrication of multi-layered bipolar ionic diode (**A**). Soft-lithography process to fabricate multi-layered bipolar ionic diode by using negative photoresist. SU-8 2005: target 5 µm height (1st shallow channel), SU-8 2050: target 25 µm height (2nd shallow channel), and SU-8 3050: target 100 µm height. Deep channel (**B**). Fabrication process for CS-NCNM and AS-NCNM by using self-assembly of negatively charged and positively charged nanoparticles.

**Figure 2 micromachines-14-01311-f002:**
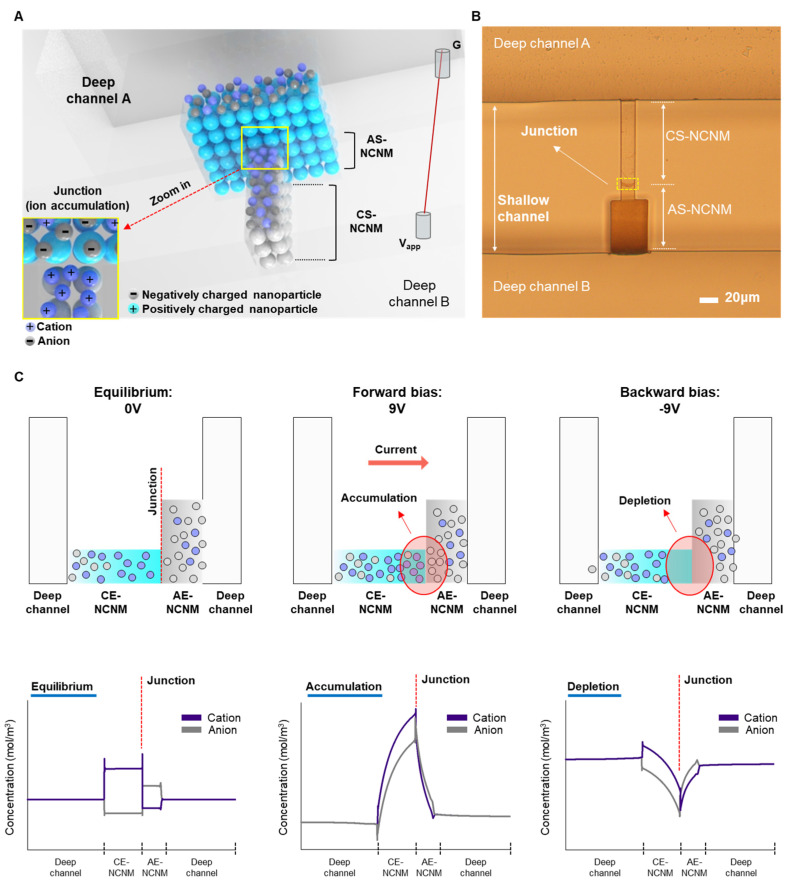
Schematic, fabrication result, and working principle of the proposed multi-layered bipolar ionic diode. (**A**) Schematic of multi-layered bipolar ionic diode. (**B**) Inverted microscopic image of the fabricated multi-layered bipolar ionic diode. (top view). (**C**) Working principle of multi-layered bipolar ionic diode that presents the different ionic distributions depending on the direction of the applied potential (side view).

**Figure 3 micromachines-14-01311-f003:**
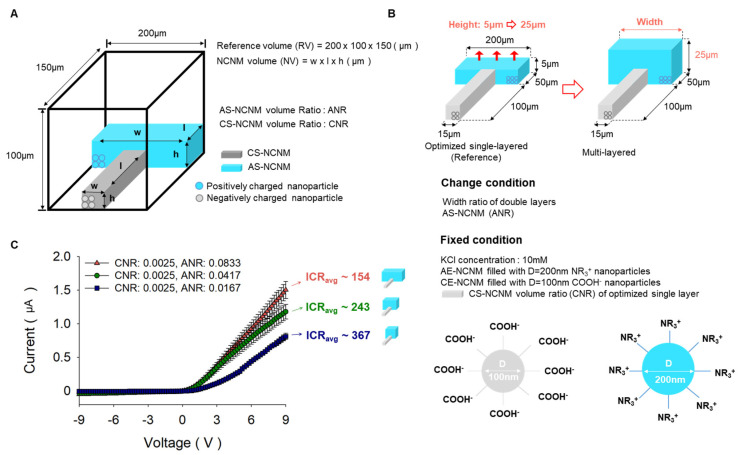
Investigation of the ICR value for multi-layered bipolar ionic diode according to the change of geometry. (**A**) Definition of AS-NCNM and CS-NCNM volume ratio. (**B**) Experiment condition of nanoparticles, KCl concentration, and geometry change for multi-layered bipolar ionic diode. (**C**) I–V curve results for multi-layered bipolar ionic diode according to the change of width in AS-NCNM.

**Figure 4 micromachines-14-01311-f004:**
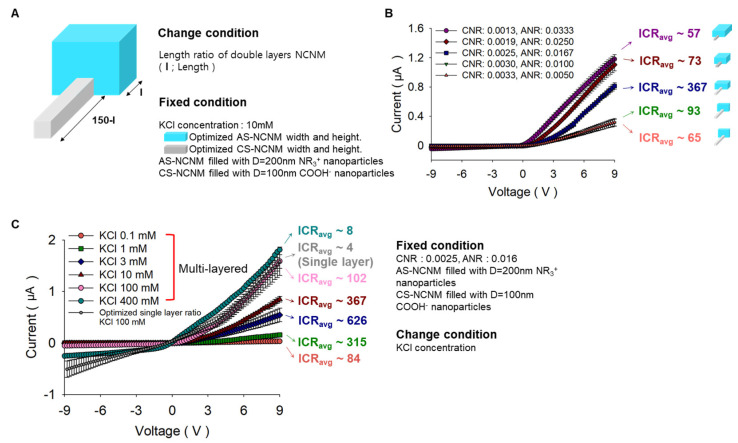
Exploration of ICR characteristics according to the change of length ratio and various salt concentrations. (**A**) Experiment conditions for KCl concentration, nanoparticles, and length ratio of AS-NCNM and CS-NCNM of the multi-layered bipolar ionic diode. (**B**) I–V curve of ICR results with respect to the length ratio of AS-NCNM and CS-NCNM of the multi-layered bipolar ionic diode. (**C**) I–V curve of ICR results for various KCl concentrations.

**Figure 5 micromachines-14-01311-f005:**
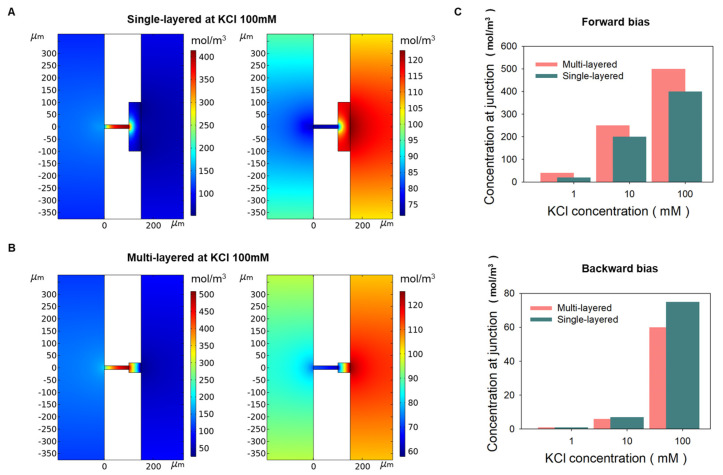
Numerical multi-physics simulation for the multi-layered and single-layered bipolar ionic diode (**A**) When forward and backward bias was applied at KCl 100 mM, concentration at junction on 2D single-layered simulation model (top view). (**B**) When forward and backward bias were applied at KCl 100 mM, concentration at junction on 2D multi-layered simulation model (top view). (**C**) The concentration of forward bias and backward bias at junction at multi-layered and single-layered when KCl 1, 10, and 100 mM.

**Figure 6 micromachines-14-01311-f006:**
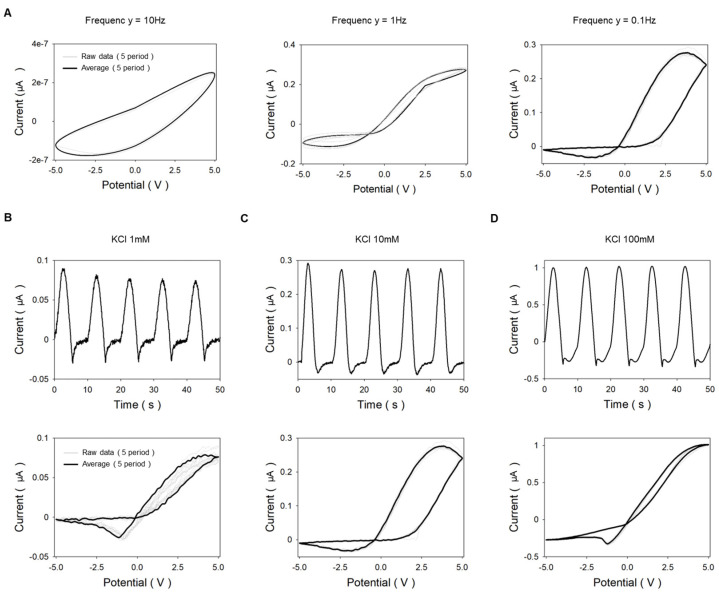
Current–voltage diagram and current–time curve of multi-layered bipolar ionic diode under AC potential. (**A**) Current–voltage diagram at a KCl concentration of 10 mM according to various frequencies. (Gray line: 5 period data, bold black line: Average of 5 period data). (**B**) Current–voltage diagram and current–time curve at KCl 1 mM (low concentration). (**C**) Current–voltage diagram and current–time curve at KCl 10 mM (nominal concentration). (**D**) Current–voltage diagram and current–time curve at KCl 100 mM (high concentration).

**Table 1 micromachines-14-01311-t001:** Comparison of ion current rectification of other works with proposed device in various ionic concentrations.

Author, [Ref] Year	Material, Device Fabrication Method	Voltage Bias	ICR in 0.1 mM and 1 mM	10 mM	100 mM
Zhang et al. [19] 2015	Poly(ethylene terephthalate) (PET)/track etching technique	From −2 to 2 V	0.1 mM~100 1 mM~430	~1000	~200
Li et al. [20] 2019	Polycarbonate (PC)/track etching technique	From −1 to 1 V	0.1 mM~5 1 mM~7	~100	~650
Zhao et al. [21] 2020	Poly(ethylene terephthalate) (PET)/track etching technique	From −2 to 2 V	1 mM~10	~200	~250
Liu et al. [22] 2020	Anodic aluminum oxide (AAO)/anodization	From −2 to 2 V	0.1 mM~20 1 mM~80	~500	~100
Liu et al. [23] 2020	Polyacrylic acid (PAA)/anodization	From −1 to 1 V	0.1 mM~300 1 mM~430	~160	~30
Kim et al. [36] 2022	Polydimethylsiloxane (PDMS)/soft lithography	From −9 to 9 V	0.1 mM~50 1 mM~200	~370	~4
This study	Polydimethylsiloxane (PDMS)/soft lithography	From −9 to 9 V	0.1 mM~85 1 mM~320	~370	~100

## Data Availability

The data presented in this article are available on request from the corresponding author.

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
