# Peer review of "Multi-Layered Bipolar Ionic Diode Working in Broad Range Ion Concentration"

_micromachines, 2023, doi:10.3390/mi14071311_

Round 1

Reviewer 1 Report

In the present study, the authors proposed a multi-layered bipolar ionic diode based on an asymmetric nanochannel network membrane (NCNM), which is realized by soft lithography and self-assembly of homogenous-sized nanoparticles. Both simulation and experimental results are provided in this interesting paper, and the analyzes and discussions support well the conclusion. This manuscript is of high quality and the referee here suggested its acceptance for publication after doing a minor revision:

1. Line 70, wad→was.

2. In Eq.(2), the gradient of ion concentration is wrongly missed in the diffusion term.

3. Line 127 and 129, surface charge density→volumetric free charge density.

4. Fluid motion caused by electrostatic force acting on the free charge density (Eq.5) is not considered by this work, please explain why electroosmotic convection of charges can be ignored.

5. Some recent literatures on nanofluidic ion current rectification should be mentioned and cited in current work, e.g. [Electrochimica Acta, 2022, 403: 139706][ ACS nano, 2022, 16(3): 4930-4939.][Physics of Fluids, 2017, 29, 112001].

6. What kind of mesh have the authors used in the simulation?

Reviewer 2 Report

The manuscript discusses the significance of ion current rectification (ICR) and presents a novel multi-layered bipolar ionic diode that demonstrates enhanced performance across a wide range of ion concentrations. I think the manuscript topic is of interest for special issue of “Micro/Nanostructures in Sensors and Actuators”. However, the current version of the manuscript requires substantial revisions to meet the standards for publication. The authors offer a work that potentially can improve our knowledge about the topic of interest. I suggest that the authors consider a revision of their work along the following suggestions and questions.

1. Abstracts must include the most prominent results for the work to be  valuable, which are currently lacking. It is suggested to discuss more about the findings of this study in the abstract.

2. In the introduction, I would suggest clearly discussing the different shapes of
nanochannels. Why are the shapes that have been chosen relevant? Where can they be found, and what are their applications? As much work has already been done in this research field, it is essential to emphasize this numerical study's novelty and potential relevance in the manuscript. To this end, please see the relevant references such as:

https://doi.org/10.1016/j.nanoen.2020.104612
https://doi.org/10.1016/j.electacta.2021.139376
https://doi.org/10.1016/j.electacta.2018.10.074
https://doi.org/10.1016/j.electacta.2022.141175

3. I strongly recommend that the authors include a paragraph in their manuscript discussing the unique contributions and novelty of their work in comparison to previous studies in the literature. This addition will effectively highlight the distinguishing features of their research. I suggest revising both the abstract and introduction sections to incorporate this discussion and emphasize the novelty. By doing so, the authors will motivate readers of the "Special Issue" to engage with their work.

4. How does the performance of the proposed multi-layered bipolar ionic diode compare to previous attempts in improving ion current rectification? For more information, refer to:

https://doi.org/10.1002/adfm.201801079
https://doi.org/10.1016/j.molliq.2021.118324
https://doi.org/10.1088/0957-4484/21/26/265301

5. The authors should provide more details about the specific characteristics of the nanochannel network membrane (NCNM) and on how it contributes to the overall performance of the diode.

6. Has the proposed diode been tested or demonstrated in any practical applications related to biosensors, desalination, or energy harvesting? If so, what were the results or potential benefits observed?

7. The following relevant references must be cited properly in the introduction and throughout the manuscript to address the mentioned topics:

https://doi.org/10.1021/acs.langmuir.2c01790
https://doi.org/10.1038/s41467-022-32590-9
https://doi.org/10.1021/nl800949k
https://doi.org/10.1039/D2CP01015A
https://doi.org/10.1016/j.electacta.2021.139221
https://doi.org/10.1039/D0CP05974A
https://doi.org/10.1002/admt.202000765

8. Critical commentary is needed from the author who eventually recommends directions for further research.

9. What are the limitations of this study? I recommend the authors to highlight this topic.

10. Please edit the language carefully, fix typos, and correct grammatical errors. Also, the layout of the references should be thoroughly revised. Please see the newly published papers in Micromachines.

Reviewer 3 Report

The paper “Multi-layered bipolar ionic diode working in broad range ion concentration” describes the continuation and improvements from previous results (ACS Nano 2022, 16, 8253-8263) using single-layered bipolar ionic diode with optimization of nanochannel network membrane (NCNM) region. In the previous publication, the author obtained the highest ionic current rectification (ICR) degree (~1600) only at a narrow concentration (1 – 10 mM) but significantly reduced at lower (<0.1 mM) and higher (>100 mM) concentration of electrolyte. In this work, the author claimed that this developed platform could obtain a more stable performance of ICR at broader concentration of electrolyte. This result is intriguing and interesting, and I would like to suggest for the publication. However, I have found several questions regarding the manuscript as follows:

1.       Why the author chooses the word “multi-layered” as their title? because in this work, the authors have fabricated several bipolar ionic diodes with different heights of nanochannel network membrane (NCNM)

2.       In the materials and methods section (Figure 1A), how can we know the fabrication of shallow channel and deep channel in ionic diode device?

3.       In the materials and methods section (line 101), is there any information about what kind of nanoparticles used to fabricate cations selective (CS) and anion selective (AS)-NCNM?

4.       In the materials and methods section (Line 105), does it mean platinum (Pt) electrodes for PT electrodes?

5.       In the results and discussion section (line 141), why in Fig. 2B from the image of “inverted” microscope? While in Figure caption 2B (line 154), it is just the microscopic image.

6.       In the results and discussion section (line 152), is there any information about the typical experimental environments used to these 3 figures (Fig. 2C in the bottom row)?

7.       In the results and discussion section (line 143), why the bipolar diode displays the preferable ionic distributions in respect to their surface charge polarity?

8.       In the results and discussion section (Fig. 3A), is there any information about width (w), length (l), and height (h) used to fabricate CS-NCNM?

The quality of English is fair and can be well-understood 
